

# Hydrometeorological evaluation of two nowcasting systems for Mediterranean heavy precipitation events with operational considerations

Alexane Lovat[1], Béatrice Vincendon[1], and Véronique Ducrocq[2]

[1]DSO Météo France, 42 av. G. Coriolis, 31057 Toulouse cedex 1, France
[2]DIROP Météo France, 42 av. G. Coriolis, 31057 Toulouse cedex 1, France

**Correspondence:** Alexane Lovat (alexane.lovat@meteo.fr)

**Abstract.** Heavy precipitation events and subsequent flash floods regularly affect the Mediterranean coastal regions. In these situations, forecasting rainfall and river discharges is crucial especially up to six hours, which is a relevant lead time for emergency services in crisis time. The present study investigates the hydrometeorological skills of two new nowcasting systems: a numerical weather model AROME-NWC and a nowcasting system blending numerical weather prediction and extrapolation of radar estimation called PIAF. Their performance is assessed for 10 past heavy precipitation events that occured in southeastern France. Precipitation forecasts are evaluated at a 15 min time resolution and the availability times of forecasts, based on the operational Météo-France suites, are taken into account when performing the evaluation. Rainfall observations and forecasts were first compared using a point-to-point approach. Then the evaluation was conducted from an hydrologic point of view, by comparing observed and forecast precipitation over watersheds affected by floods. In general, the results led to the same conclusions for both evaluations. On the very first lead times, up to 1h15/1h30 of forecast, the performance of PIAF is higher than AROME-NWC. For longer lead times (up to 3h) their performance are equivalent in general. An assessment of river discharges simulated with the ISBA-TOP coupled system, which is dedicated to Mediterranean flash-flood simulations, forced by AROME-NWC and PIAF rainfall forecasts, was also performed on two exceptional past flash flood events. The results obtained for these two events show that using AROME-NWC or PIAF rainfall forecasts is promising for flash-flood forecasting in terms of peak intensity, timing, and first rise of discharge, with an anticipation of these phenomena that can reach several hours.

## 1 Introduction

The Mediterranean coastal regions are frequently affected by heavy precipitation events (Ricard et al., 2012). The mesoscale convective systems associated with these events produce a large amount of rainfall, typically greater than 100 or 200 mm in only a few hours. The occurrence of such intense rainfall over small areas and catchments up to a few hundreds of square





kilometers often trigger devastating flash floods. These events threaten people as well as property (Drobinski et al., 2014; Gaume et al., 2016) and result in direct economic losses valued at hundreds of millions of euros each year. Even if significant progress has been realised last decades, very localised and high intensity rainfall events such as those generated with the

Mediterranean precipitating systems are difficult to forecast (Alfieri et al., 2012; Vié, 2012; Silvestro et al., 2017). Among the difficulties encountered to forecast heavy precipitation events with accurate intensity, chronology and location, the most important are the complex features and variability of deep convection and the associated small space-time scales that are hardly predictable. Nowcasting systems suit these scales with high spatial and temporal resolution short-term forecasts (usually up to a few hours). They can be based on extrapolation of observation, or rely on mesoscale numerical weather prediction or

else combine these two approaches. The concept of extrapolation of radar echoes for short-term precipitation forecasting was first developed in 1953 (Ligda, 1953). This approach has been deepened by the search for fast-computing algorithms or methods allowing the best forecasts of the evolution of precipitating cells (in terms of displacement, size and intensity) from the radar imagery data. Among them there are those based on cross-correlation (eg Rinehart and Garvey, 1978) and those based on individual radar echo-tracking (Johnson et al., 1998; Dixon and Wiener, 1993). These methods were later improved

to ensure the consistency of the velocity field reconstructed after extrapolation (Li et al., 1995; Laroche and Zawadzki, 1995; Germann and Zawadzki, 2004). Although widely used, the accuracy of extrapolation methods is limited because they are not able to forecast convective storm initiation, growth and decay (Golding, 1998). Beyond one or two hours, the use of numerical weather prediction is necessary to depict rapidly changing conditions. The latest generation of mesoscale models allows to reach kilometer horizontal resolutions and is able to reproduce fine-scale boundary layer and convection processes. Numerical

weather prediction systems have been configured to meet the requirements of nowcasting, i.e. to have very short-term forecasts, updated with the latest observations, in the fastest possible time. These systems are based on the same kilometric horizontal resolution models as the forecast models used for forecasting the weather over the next 24-48 hours, but their assimilation frequency and windows are adapted to allow the forecasts to be refreshed frequently with new observations while ensuring short forecast delivery times (Weygandt et al., 2009). AROME-NWC (Auger et al., 2015), which is a system operationnaly

used at the French meteorological service Météo-France, is one of them. Thus, several nowcasting system types with various skills coexist at lead times between 1 and 3-4 hours. Methods have been developed to combine extrapolation methods and numerical prediction systems. Seamless forecasts can be obtained by weighting precipitation fields from radar extrapolation and numerical weather prediction forecasts. The first approach in blending nowcasting was introduced by Golding (1998) with an heavy weight for the extrapolation nowcasting during the first hour and the heavier weights transitioning to the numerical

forecasts with increasing lead time. A new nowcasting system called PIAF (Moisselin et al., 2019) blending numerical weather prediction and extrapolation of radar estimation has recently been developed by the nowcasting department at Météo-France. Its skills for rainfall forecasting still need to be assessed.

The short hydrological response times ranging from few minutes to few hours after heavy downpoors are a major issue for notifying at-risk populations and planning the intervention of emergency services in crisis time. The use of rainfall nowcasting

allows to extend the lead time of hydrological forecasts by a few hours compared to the mere use of observed precipitation data that limits the forecast lead time by the catchment response time.Vivoni et al. (2006), Berenguer et al. (2005) and Dolcine et al.

(2001) have demonstrated the benefit of using deterministic rainfall forecasts obtained from radar-based extrapolation as input to hydrological models. Other studies explored the potential of probabilistic rainfall nowcasting for flash-flood forecasting (Silvestro and Rebora, 2012; Poletti et al., 2019). Nowcasting flash floods provides an anticipation time sufficient enough to boost the preparedness of people and civil protection and sometimes a valuable time to prevent the authorities from being completely unprepared for the occurring or upcoming event (Silvestro et al., 2017). The availability time of the rainfall forecast is thus crucial for real-time streamflow forecasting. Whereas operational hydrological models are often fastrunning (i.e. finishing in seconds to minutes) weather forecasts require more time to be delivered. Therefore, this delay is taken into account in this study to consider the operational real time constraints.

The present study investigates the hydrometeorological skill of AROME-NWC (Auger et al., 2015) and PIAF (Moisselin et al., 2019), two new nowcasting systems operationally used at the French meteorological service Météo-France. The main objectives are to compare and evaluate their performance for 10 past Mediterranean heavy precipitation events and to suggest some ”best practices” for real-time forecasting. The effect of spatial resolution, lead time and precipitation intensity on forecast skill is studied at a 15 min time resolution. Precipitation forecasts are first evaluated from a meteorological perspective with a synoptic-scale verification using a point-to-point approach. Rainfall observations and forecasts are compared at each grid point of an area covering southeastern France. Then an evaluation on scales relevant to hydrology is performed, by comparing observed and forecast rainfall averaged over watersheds affected by past floods. An assessment of river discharges simulated with ISBA-TOP, a model dedicated to Mediterranean flash-flood simulations (Bouilloud et al., 2010; Vincendon et al., 2016), forced by AROME-NWC and PIAF rainfall forecasts was also conducted on two French exceptional recent flash flood events. The performance of the forecasts is assessed in terms of intensity and timing of the flood peak.

This paper is organized as follows. Section 2 describes the case studies, the nowcasting systems, the hydrological system, and the evaluation methods. The results of the different verifications are presented and discussed in Section 3. The conclusions are reported in Section 4.

## 2 Materials and methods

### 2.1 Case studies

The selected case studies concern recent heavy precipitation events which occured in South-East France between October 2015 and November 2018. A rectangular verification zone was defined to investigate the performance of the nowcasting systems during these events. It encompasses the regions along the Mediterranean coast most favorable to intense events (black rectangle in Figure 1) such as the eastern Pyrenees, the southern Alps, and the Cévennes-Vivarais region. This area is characterized by a pronounced topography with steep slopes and narrow valleys. Within this large zone ($110000\ \mathrm{km}^2$), 19 catchments with areas ranging from 19 to $1100\ \mathrm{km}^2$ and short response times were selected for the hydrological evaluation (Table 1).Watersheds numbered 1 to 8 in Figure 1 are all tributaries of the Aude river in the North-East of the Pyrenees. Watersheds numbered 9 to 12 are located in the Cévennes region. The other watersheds numbered 13 to 19 are located in the French Riviera.





To assess the performance of the nowcasting systems, 10 recent heavy precipitation events were considered (first and sec-
ond columns of Table 2). These rainy episodes are representative of the variety of rainfall intensities and durations and the
hydrological responses of the rivers encountered in the French Mediterranean coastal regions. In particular, the two events that
occurred on 3 October 2015 in the French Riviera and 14-15 October 2018 in the Aude region are among the latest major tragic
flash-flood events that have affected metropolitan France. They represent together 34 deaths and more than 800 million euros
of damage. More details about the October 2015 event and the October 2018 event were respectively given by Payrastre et al.
(2016) and Caumont et al. (2020).

## 2.2 The nowcasting systems

### 2.2.1 AROME-NWC

AROME-NWC (Auger et al., 2015) is a configuration of the French numerical weather prediction system AROME-France
(Seity et al., 2011; Brousseau et al., 2016) especially designed for nowcasting purposes. AROME-NWC is a mesoscale and non-
hydrostatic model. Its horizontal resolution is of $1.3\,\mathrm{km} \times 1.3\,\mathrm{km}$ and its vertical grid has 90 levels ranging from 10 to around
30000m above the ground. The deep convection is explicitly resolved and the microphysical processes are governed by the
ICE3 one-moment bulk microphysical scheme (Pinty and Jabouille, 1998). AROME-NWC is thus able to forecast mesoscale
convective systems that caused heavy rain in the Mediterranean area. Boundary conditions are provided by the analysis of
ARPEGE global operational numerical weather prediction model. The AROME NWC initial conditions are provided by a
three-dimensional variational (3D-Var) data assimilation of observations available within a 20 minutes windows centered on the
analysis time, each hour. The observations are primarily radar (reflectivity and radial velocity) data, screen level measurements,
and to a lesser extent, aircraft, sounding and satellite data. Arome-NWC is run every hour and provides short-range forecasts
up to 6 hours with a time step of 15 minutes on a domain covering France and adjacent areas. Forecasts were available within
35 minutes at the time of the study.

### 2.2.2 PIAF

Very recently the nowcasting department of Météo-France has developed a new nowcasting system PIAF (for "Prévision
Immédiate Agrégée Fusionnée" in French, Moisselin et al., 2019) which is a data fusion product between radar extrapolation
and numerical prediction (the rainfall forecast by AROME-NWC). For the extrapolation of radar quantitative precipitation
estimation, radar data are proccessed as follows: rainy cells are identified by windows surrounding areas of connected pixels
above a given threshold, the displacement of each cell is determined using the previous image (highest correlation), a gridded
motion field is computed from the movement vectors of the cells with different threshold values and applied to the cells to
extrapolate them in the future.

PIAF is based on a sequential aggregation of these two predictors (radar extrapolation and numerical prediction) and the
results of blending is a linear compound of both. Its aim is to perform better than the best predictor (by minimizing the regret,
see for example Auer et al., 2002). The weights given to each predictor are adjusted according to their deviation from the



previous 6 hours observations . The weights depend also on the forecast range and on the geographical area, according to a division of France into six sub-areas. PIAF is run every 5 minutes with a 3 hours lead time and a time step of 5 minutes. Forecasts are available within 2 minutes. The Gerrity score (Gerrity Jr, 1992) is used to estimate the loss of each product with respect to the radar quantitative precipitation estimates.

## 2.3   The hydrometeorological model ISBA-TOP

ISBA-TOP (Bouilloud et al., 2010; Vincendon et al., 2016) is a distributed model designed to simulate the hydrological response of Mediterranean catchments during heavy precipitation events. It is based on the coupling between the land surface model ISBA (Interaction Surface Biosphere Atmosphere, Noilhan and Planton, 1989) and TOPODYN (Pellarin et al., 2002), a variant dedicated to flash-flood modelling of the hydrological model TOPMODEL (Beven and Kirkby, 1979). This coupling consists in introducing into ISBA a lateral distribution of soil water following TOPODYN concept. ISBA deals with the water and energy budgets within the soil column and between the vegetation and the atmosphere above. Fluxes are computed for all grid meshes of its domain. From the resulting volumetric water content, water-storage deficit is computed by TOPODYN for each watershed pixel of 50-m x 50-m resolution. TOPODYN manages the computation of the lateral redistribution of water within the catchment by using topographical indexes and the spatial variability of the rainfall. From the new saturated areas and new soil moisture fields obtained, water contents are updated in ISBA. From them, ISBA computes sub-surface runoff and deep drainage which are routed up to the river and total discharges are then produced at catchment outlets.

The ISBA-TOP configuration used in this study is the one suggested by Lovat et al. (2019) for flash-flood simulations, based on SRTM data for orography, Land use/cover area frame statistical survey topsoil data (Ballabio et al., 2016) for soil texture, ECOCLIMAP-II (Masson et al., 2003; Faroux et al., 2013) data for land cover, and a spatial resolution of 300m for ISBA.

The ISBA-TOP coupled system has been run during the HyMex special observing periods for real-time prediction of discharges for watersheds in the Cévennes-Vivarais region and the Mediterranean coastline of southeastern France. Its performance was also assessed for Italian watersheds (Nuissier et al., 2016). ISBA-TOP is used in operations in the National Institute of Meteorology and Hydrology of Bulgaria for the Arda river flood forecasting (Artinyan et al., 2016). In addition to simulating river discharges during flash floods, ISBA-TOP is also able to simulate intense runoff phenomena (Vincendon et al., 2016; Lovat et al., 2019).

### 2.4   Evaluation methods

To evaluate the quality of the precipitation forecast provided by AROME-NWC and PIAF, the $1 \, \mathrm{km}^2$ quantitative precipitation estimates ANTILOPE (Laurantin, 2008) at a 15 min time resolution, which merged observations from the Météo-France radar and the rain gauge network, were used as reference data, called observed data or observation hereafter. Two verification methods were applied for the rainfall nowcasting evaluation process. The first method, commonly used in the meteorological community, was based on point-to-point comparisons of the forecasts and observations. Comparisons were performed at each grid point of a common one-kilometre resolution grid over a large area of $110000 \, \mathrm{km}^2$ covering southeastern France (see the black rectangle of the Figure 1). The rainfall fields were downscaled over the common grid by using a nearest-grid-point interpolation





method. The second evaluation was carried out from a hydrological point of view, by comparing observed and forecast rainfall
averaged over the surface of watersheds affected by floods. For both evaluations, the available forecasts covering 10 recent
heavy precipitation events occurring in southeastern France between 2015 and 2018 were considered (Table 2).

The observations and forecasts used in this study have different time resolutions (15 min for ANTILOPE and AROME-
NWC, and 5 min for PIAF). The comparisons have been carried out by using a common accumulation time step of 15 min.
This time step allows to characterize the high rainfall temporal variability, notably for convective situations.

One original feature of this study is that the availability times of forecasts, based on the operational Météo-France suites, are
taken into account when performing the evaluation. Forecasts are released with a time delay of 35 min for AROME-NWC and
of 2 min for PIAF. To coincide and easily compare observations and forecasts over full 15 min time intervals, the first 45 min
of the AROME-NWC forecast are not considered for the evaluation. Similarly, the first 5, 10 or 15 min of the PIAF forecast
are not used in the evaluation (Figure 2).

Even if watershed integration allows to evaluate quantitative precipitation forecasts over given surfaces, it does not allow to
evaluate whether rainfall amounts occur at the right location with the right chronology within the catchments. An assessment
of AROME-NWC and PIAF performance for flash-flood nowcasting was also performed through running of hydrological
forecasts using ISBA-TOP for two events, occurred on 3 October 2015 and on 14-15 October 2018. The reference is the
discharge simulation obtained using the radar rainfall estimates ANTILOPE as input to the distributed hydrological model.
Initial conditions (surface soil water and temperature) come from the Météo-France hydrometeorological operational system
SAFRAN-ISBA-MODCOU (Habets et al., 1999). ISBA-TOP was calibrated for hourly rainfall estimates, thus only hourly
discharges were simulated. These hydrological simulations systematically stop at the end time of the rainfall forecast, even if
the basin concentration time would make it possible to extend the hydrological forecast beyond the rainfall forecast horizon.

## 3   Results

### 3.1   Point-to-point evaluation of the rainfall nowcasting

The results of the evaluation are presented for AROME-NWC and PIAF separately before comparing both systems. Scores
used here are described in Appendix A.

The mean and root mean square errors are shown as a function of lead time in Figure 3 for 15-min rain accumulation
forecasts. In general the AROME-NWC root mean square error increases slightly with lead time. The mean error resulting
from the difference between forecasts and observations, is negative indicating that precipitations are underestimated by the
model on average. Four standard categorical verification scores are presented in Figure 4 to summarize the performance of
15-min rain accumulation forecasts from AROME-NWC and PIAF. These are the hit rate, the false alarm rate, the Heidke
Skill Score (HSS) and the frequency bias as functions of lead time for two thresholds characterising precipitation occurence
and more intense precipitation: 0.5mm/15min and 3mm/15min. The hit rate and HSS slightly decrease with increasing lead
time. The false alarm rate of AROME-NWC depends little on the forecast lead time regardless of the threshold. For the higher





rainfall intensities (3mm/15min) the frequency bias is greater than 1, indicating a trend to predict too frequently these rainfall accumulations at all lead time.

The PIAF root mean square error increases with lead time up to 2 hours and decreases very slightly beyond (Figure 3b). During the very early forecast period it increases very quickly. This can be explained by the quality of the extrapolation of radar data, which deteriorates quickly with the lead time. Indeed the effects of advection errors accumulate and increase with successive time steps. The mean error for PIAF is also negative, indicating an underestimation of the precipitation amount on average (Figure 3a). A quick loss of PIAF accuracy is observed in Figure 4 (decrease in hit rate and HSS) in the first hour of lead time. It is still decreasing up to 1h40 and becomes stable thereafter. As well as for AROME-NWC, too many high rainfall accumulations (3mm/15min) are forecast by PIAF (Figure 4d).

Comparison of the skills of the two nowcasting systems reveals that for lead times in the range T+2 hours to T+3 hours, AROME-NWC obtains on average better results than PIAF. For lead times in the range T+1 hour to T+2 hours, the performance of the two forecasting systems is often close. At lead times less than 90 minutes or 75 minutes, depending on the intensity of the rainfall forecast, the performance of PIAF generally exceeds that of AROME-NWC.

PIAF results from the linear combination of AROME-NWC prediction fields and radar extrapolation. The weights given to each predictor are adjusted according to their recent performance against observation. In general an important weight is given to the extrapolation in the first time steps and for longer forecast times the numerical prediction gains more importance to the point that the rainfall field is entirely provided by AROME-NWC at the end of the PIAF forecast. Note for example that for the latest PIAF lead times, the root mean square errors converge to AROME-NWC errors beyond 2.5 hours (Figure 3b). However, the quality of PIAF and AROME-NWC may not be equivalent for the same forecast horizon. Indeed PIAF forecasts are systematically based on the latest available AROME-NWC run and therefore the forecast lead time for the AROME-NWC run used in PIAF may be older than that indicated in the PIAF assessment. Since the AROME-NWC availability time is 35 minutes, for an AROME-NWC run initiated at round hour H, only PIAF forecasts initiated between H+40 minutes and H+95 minutes will use this run, and those between H and H+35 minutes will use the AROME-NWC run launched at H-1. Thus the same lead time of two PIAF forecasts does not necessarily rely on the same lead time of the two associated AROME-NWC runs. Differences between PIAF and AROME-NWC skills at the last PIAF forecast lead times may also be explained in several cases, with the fact that PIAF does not switch completely to AROME-NWC (e.g. cases where AROME-NWC is too far away from observations over the last six hours).

The main drawback of the point-to-point verification, especially in the case of convective situations, is to give significant weight to even small location errors (the so-called "double penalty", Anthes, 1983; Gilleland et al., 2009). To give more credit to "close" forecasts a fuzzy method was used to measure the similarity between forecasts and observations in local neighborhoods of the observations. Fraction Skill Scores (FSS, Roberts and Lean, 2008) were applied to compare forecast and observed coverage of rain exceeding certain thresholds in spatial windows of increasing size. The FSS were also used to compare AROME-NWC and PIAF and to analyse their performance with the forecast range. The five neighborhood scales used are 1, 5, 10, 20 et 40 km, and six 15-min precipitation thresholds of 0.5, 1, 2, 3, 5 et 10 mm were selected. Figure 5 and Figure 6





show respectively for AROME-NWC and PIAF the 10-events mean FSS results for the various thresholds and window sizes at each forecast lead time. As might have been expected, the greatest skill (highest FSS values) is associated with the largest window and the smallest threshold while the lowest skill (FSS values near 0) is associated with the smaller spatial window and largest threshold. For AROME-NWC, at all lead time the FSS monotically increases with the increase in spatial scale. For
PIAF, there is a rapid and significant decrease in FSS values with the forecast range, up to 1h45 lead time. It is mainly due to the decrease in the quality of the extrapolation forecast with time. The FSS computed from the PIAF precipitation forecast are generally better than those from AROME-NWC, at least for the first 90 minutes of forecast lead time.

For verifying AROME-NWC and PIAF forecast performance, traditional verification method focused on point-to-point statistical features was used as well as the neighborhood spatial technique FSS. Based on their similar results, it can be summarized
that a quick loss of PIAF accuracy is observed on the very first lead times, its performance is higher than AROME-NWC up to 1h15/1h30 of forecast but not necessarily beyond.

### 3.2 Evaluation of the rainfall nowcasting at the catchment scale

To complete and verify the conclusions of the point-to-point evaluation, the rainfall forecasts averaged over the catchments were also studied. Indeed the amount and the location of rainfall forecasts at the catchment scale are essential for hydrological
response forecast (Yates et al., 2006; Anquetin et al., 2005). The studied watersheds are those specified in Table 2.

Just as before, the mean and root mean square errors over watersheds are shown as a function of lead time in Figure 7 for 15-min cumulated rainfall. The root mean square errors for AROME-NWC increase with the lead time. This increase is not strictly monotonous and is noisy due to the size of the sample: for example, a slight improvement in scores for AROME-NWC can be seen for lead times between 165 minutes and 210 minutes (Figure 7b). At lead times up to 4 hours and 15 minutes,
AROME-NWC underestimates on average the rainfall accumulations (Figure 7a). The largest overestimates are observed for the events of 2015 and mid-October 2018, as shown by the inter-quartile ranges in Figure 8, which represents the forecast error distributions (forecast values - observed values) of AROME-NWC and PIAF. The hit rate, the false alarm and the HSS vary little with the forecast lead time for the lowest threshold (0.5mm/15min) whereas the signal is more noisy for the 3mm/15min threshold with the hit rate and the HSS deteriorated with the lead time (blue markers for AROME-NWC in Figure 9). For the
higher rainfall intensities, the frequency bias increases with the lead time and is significantly greater than 1, indicating that AROME-NWC over-predicts these rainfall intensities. There is a loss of PIAF forecast accuracy with increasing forecast lead time (Figure 7b). As seen previously for the point-to-point evaluation, the root mean square errors increase very rapidly over the first hour of the forecast.

On average, PIAF rain accumulation forecasts are lower than those observed. The most significant forecast errors are due to
the event of October 3, 2015 and more particularly on the Brague river at Biot where the extremes of error in 15-minute rainfall forecasts are approximately +/-20 mm. In terms of categorical scores (green markers for PIAF in Figure 9), there is a visible decrease in the hit rate and HSS as a function of the forecast lead time for the two rainfall thresholds studied. The frequency bias is close to 1 for the 0.5mm/15min threshold and greater than 1 for the higher threshold.





Finally, the results of both evaluation methods in general lead to the same conclusions. The performance of PIAF is very

good to good over the first hour of forecasting, but it deteriorates very quickly, to reach about the same or even a lower skill than AROME-NWC beyond about 1h15/1h30 of forecasting. Between 2 and 3 hours of forecasting, AROME-NWC performs better or at the same level as PIAF. It is worth mentionning that the values of the scores as a function of lead time show more variability from one lead time to the next compared to those of the point-to-point evaluation. This might be due to a smaller size of the evaluation sample.

**3.3 Hydrological evaluation for two case studies**

The potential of AROME-NWC and PIAF for flash-flood nowcasting is introduced through the running of hydrological simulations using ISBA-TOP for two French major events occurred on 3 October 2015 and on 14-15 October 2018 (see for example Figure 10). These two remarkable events, which are part of the sample used for the rainfall assessment were selected for their significant hydrometeorological characteristics with very intense precipitation and river overflows.The evaluation aims at

addressing the following question: How many hours of anticipation on floods can we have at most in terms of intensity and temporality of the flood peaks using rainfall nowcasting ? From all the hydrological forecasts provided by ISBA-TOP driven by the rainfall forecast of October 2015 and 2018 (Table 2), the best anticipation of three phenomena was studied per watershed. These are:

- the start of increased discharge, defined here as an increase of at least $5 \text{ m}^3\text{s}^{-1}$ in one hour.

- the right order of magnitude of the peak flow value (meaning an error of less than 30% with respect to the reference peak discharge).

- the peak time which is also the start of the recession limb.

For a given discharge forecast, the anticipation of one of these phenomena is calculated as the duration between two moments in time: the starting time of the rainfall forecast and the time of the phenomenon in the reference hydrograph (rainfall estimates

as input to ISBA-TOP). Results are presented in tables 4 and 5. The objective of these tables is to represent the behaviour of the best discharge forecasts for each watershed. They also provide a concise and comprehensive view of the results. For the "peak time" phenomenon, the anticipation period is only considered if the forecast and reference phenomenon occur at the same time. For the phenomena "start of increased discharge" and "flood peak of the right intensity", a one-hour delay between forecast and reference is accepted. A different colour is assigned depending on the error between the reference flood peak and

the forecast flood peak. If the difference between forecast and reference intensity is less than 10%, it is coloured green, if the error is between 10% and 20%, it is coloured orange and if it is between 20% and 30%, it is coloured red.

To facilitate the reader's understanding of the tables, the line summarizing simulations for the Fresquel river at Pezens obtained by ISBA-TOP driven by AROME-NWC forecasts (Table 3, coming from Table 4) is taken as an example and is now detailed. Associated discharge time series are shown in Figure 11. In the reference simulation (in black on the Figure 11), the

flow starts increasing at 23:00UTC on October 14. To calculate the anticipation on this rise, we look for the oldest run which forecasts a rise at 22:00UTC, 23:00UTC or 00:00UTC. In this case, it is the run starting at 18:00UTC, so the anticipation on the rise is 5 hours. In the column "Anticipation of the rising flow" of Table 3, we grey out 5 hours before the hour of the





phenomenon. To have an idea of the number of cases for which the flow rises are not forecast at the right time, we indicate in the same column of the table the number of runs launched before the flood start time on the reference hydrograph (before

23:00UTC here) which forecast a rise, and among them the number which forecast a rise at +/-1 hour before the reference time. Here, five runs starting before 23:00UTC forecast a rise (those from 18:00 to 22:00UTC), four of them between 22:00 and 00:00UTC (all except the run starting at 20:00UTC which forecast a rise 2 hours later). In the column "Anticipation of the rising flow" of Table 3, "5/4" is thus written. The reference peak, with an intensity of $166 \ \mathrm{m^3s^{-1}}$, is simulated at 06:00UTC on October 15. Flood peaks are simulated by ISBA-TOP driven by the AROME-NWC forecasts starting from 02:00 to 05:00UTC.

To evaluate the anticipation on peak intensity, all peaks forecast at 06:00UTC on October 15 +/- 1h are considered. Among the runs mentioned, only those starting at 04:00 and 05:00UTC simulate a flood peak with an intensity error of less than 30%. The earliest run (starting at 04:00UTC) is represented in the table: the anticipation on the right intensity of the flood peak is thus 2 hours. For this forecast starting at 04:00UTC, the difference between the forecast peak and the reference peak is less than 10%, so the 2 hours before the hour H are coloured green in the column "Anticipation of the peak value" in Table 3. To have an idea

of the number of cases where a flood peak is forecast with an intensity close (or not) to the reference one, a triplet of values separated by slashes is indicated in the same column. These values correspond in the following order: to the total number of forecasts which forecast a flood peak, to the number of forecasts which forecast a flood peak at +/-1h of the reference peak and to the number of forecasts which forecast a flood peak at +/-1h of the reference peak and whose intensity error is less than 30%. In this case for the event of 15 October 2018, among the seventeen AROME-NWC forecasts used, four simulate a

flood peak (runs starting between 02:00 and 05:00UTC), the four present a peak timing error of at most one hour, but only two with an intensity error of less than 30% (forecasts starting at 04:00 and 05:00UTC). In the column "Anticipation of the peak value" of Table 3, "4/4/2" is written. The peak in the reference hydrograph occurs at 06:00UTC. The first forecast that triggers a recession at the right time is that of 02:00UTC, the anticipation on the peak time is therefore 4 hours. So 4 hours are grayed in the column "Anticipation of the peak timing" of Table 3. Note that the flood peak simulated with this run starting at 02:00UTC

is very overestimated. The anticipation of the right intensity of the flood peak does not always coincide with the anticipation of the time of the flood peak. If, as in this example, the duration of the anticipation of the right intensity of the flood peak is shorter than that of the peak timing, it means that older runs anticipate flood peaks but with a strong over- or underestimation. On the contrary, a duration of anticipation of the good intensity of the flood peak greater than that of the peak timing indicates that the predicted intensity of the flood peak was right at +/-1h but that the exact time of the flood peak was predicted only

later. Note that to take into account the actual availability time of the rainfall forecasts, this time should be subtracted from the anticipation time indicated in the tables.

    Among the best hydrological scenarios simulated with the AROME-NWC rainfall forecasts, the start of rising flows is anticipated at least two hours ahead, regardless of the event (Table 4). In most cases, the anticipation is more than four hours, and even, for the Orbiel and Lauquet rivers, an increase in flows is forecast up to six or seven hours in advance at the reference

peak time +/-1h . Anticipation time is not proportional to the size of the watersheds, i.e. the smaller catchments with the lowest concentration times do not necessarily have a shorter anticipation time. Among the runs forecasting a rise of discharge, there are errors in peak timing greater than one hour (for the 14/10/2018 case in particular, see in the column "Anticipation of the rising





flow" in Table 4), the time of the first rise of discharge is most often simulated too early. This is the case of the Ognon river at Pépieux where the start of the rising limb is anticipated fourteen hours in advance. Although hydrological information can

be extracted from these simulations, considering them as true flood signals could yield hits but could also entail false alarms. Among the hydrological simulations based on PIAF rainfall forecasts, the best anticipation times of the onset of increasing discharge are always greater than one hour and fifteen minutes and reach three hours and forty-five minutes on the Lauquet river at Saint-Hilaire (Table 5).

The colored cells in the column "Anticipation of the peak value" in tables 4 and 5 indicate the best anticipation time of

flood peak intensities. For the simulations based on AROME-NWC, the maximum anticipation on the flood peak with an error on the intensity lower than 10 % (green cells Table 4), ranges from one hour to four hours according to the catchment. Considering the forecast peaks for which the error on the peak intensity is higher (error between 10 and 30%) allows to gain up to one or two hours, as in the case of the Orbiel basin. With a tolerance of 20% error on the intensity of the flood peak, the anticipation is ranging from one hour to four hours. In the simulations based on PIAF forecasts, the anticipation of the correct

intensity of the flood peak (error less than 10%, green cells in Table 5) is between thirty-five minutes and two hours and fifteen minutes. Considering the anticipation on the more erroneous intensity peaks allows an additional anticipation of one hour and twenty-five minutes for the Loup river at Villeneuve-Loubet. For the simulated peaks with an intensity error of less than 30%, the possible anticipation of flood peaks with ISBA-TOP can be estimated at approximately one hour thirty/two hours, which is an order of magnitude consistent with that found for a satisfactory performance of the nowcasting system during the rainfall

forecasts evaluation.

For all the catchments, the peak times based on the AROME-NWC rainfall forecasts are anticipated with a maximum of one to five hours (column "Anticipation of the peak timing" of Table 4). Flood recession in the catchments affected by the 03/10/2015 case is systematically anticipated four or five hours in advance, this anticipation time is more variable for the 15/10/2018 event. In simulations based on PIAF rainfall forecasts, the start of the receding limb is in general forecast an hour

and a half, two hours ahead at best, except for the Orbiel river at Bouilhonnac, where the advance is only twenty minutes (Table 5).

These results obtained for two major flash-flood events are therefore indicative of the promising use of AROME-NWC or PIAF rainfall forecasts for hydrological forecasting with an anticipation of peak intensity, timing, and first rise of discharge that can reach several hours.

**4  Conclusions**

The Mediterranean regions are regularly exposed to heavy rainfall events, which can trigger devastating flash floods. In these situations, hydrometeorological forecasts up to a few hours are crucial for increasing the preparedness of the authorities and planning the intervention of emergency services. In the current study, the potential of two nowcasting systems, recently developped at Météo-France, has been assessed for forecasting Mediterranean intense rainfall events and floods. Precipitation

forecasts were evaluated in southeastern France for 10 past heavy precipitation events using a point-to-point approach and an



areal verification over watersheds affected by floods. The availability time of the rainfall forecast, which is of non-negligible time at the nowcasting ranges (few minutes to 6h) was taken into account to consider constraints of forecasters during real-time operations. These assessments for a large area in the South-East and for the catchments affected by the floods lead to the conclusion that:

- PIAF is of very good quality over the first hour of forecasting, thanks in particular to the quality of the radar extrapolation. AROME-NWC is of good quality throughout its forecast, even if its skills tend to decrease slightly with the lead time.

    - Heavy rain is predicted too often with AROME-NWC and PIAF.

    - PIAF is of higher quality than AROME-NWC on the very first lead times. This quality deteriorates very quickly, to reach a quality comparable (or even lower) than AROME-NWC beyond about 1h15/1h30 of forecast. For lead times between 2h and

3h, the performance of AROME-NWC is higher or equivalent to that of PIAF in general.

This evaluation points out the strengths and weaknesses of the nowcasting system types during extreme events, that should be considered before selecting the best method or combination for future studies or operational purposes. Depending on the accuracy of the initial radar rainfall estimation and the spatial distribution and intensity of the precipitation, radar extrapolation can provide valuable nowcasting information in the very short-term forecast. However due to dynamical evolution of precipita-

tion, especially in convective situations, there is a rapid decrease in accuracy with forecasting lead time. The above results show that blending the extrapolation of radar data with numerical prediction forecasts allows to improve the nowcasting accuracy and to extend the lead time beyond the characteristic one-hour lead time of extrapolation methods. Indeed, the use of numerical forecast helps to overcome the limitations of extrapolation at the initiation and decaying stages of convective cells and to better take into account the impact of the relief on the evolution of precipitation. The use of numerical forecasts is particularly appro-

priate for lead times greater than 2h if the numerical prediction system has been designed for nowcasting purposes. Resulting forecasts are thus frequently refreshed with new observations and are produced with a higher time resolution and a reduced calculation time compared to the time needed to run other numerical systems.

This study also highlights the benefit of the combined use of rainfall nowcasting and a distributed hydrologic model for flash-flood forecasting. It was investigated for two catastrophic French events occurred on 3 October 2015 and on 14-15 October

2018. The hydrological forecasts for small watersheds are sensitive to both the rainfall intensity and location. Nowcasting provides relevant information up to a few hours range in terms of amount, timing, and basin-specific locations of rainfall. Of the three phenomena studied (start of increased discharge, peak time and value), it is on the increase in discharge that we find in general the best anticipation times for AROME-NWC and PIAF. The peak time and intensity can be anticipated up to one to five hours in advance from the AROME-NWC forecasts. Up to one hour and a half, or two hours, PIAF allows to forecast

well peak value and time. This hydrological assessment will certainly be completed within the framework of the French PICS project (Payrastre et al., 2019) with the use of other hydrological models and the analysis of new case studies.





*Data availability.* The Shuttle Radar Topographic Mission (SRTM) 90m digital elevation data, originally produced by NASA, are available at the Consortium for Spatial Information of the Consultative Group for International Agricultural Research (CGIAR-CSI) geoportal (http://srtm.csi.cgiar.org/, last access: 28 March 2019).

**Appendix A: Verification metrics.**

The mean error (ME) measures the averaged error magnitude as follows:

$$ME = \frac{1}{N}\sum_{i=1}^{N}(f_i - o_i) \tag{A1}$$

where $o_i$ and $f_i$ are the observed and forecast value respectively and N the number of forecast-observation pairs.

It ranges from $-\infty$ to $+\infty$ and is zero if the forecast is perfect. A positive value indicates overestimation, a negative value
indicates underestimation. Note with this score, strong errors in the opposite direction can compensate each other.

The root mean square error (RMSE) is defined as follows:

$$RMSE = \sqrt{\frac{1}{N}\sum_{i=1}^{N}(f_i - o_i)^2} \tag{A2}$$

It ranges from $-\infty$ to $+\infty$ and is zero if the forecast is perfect.

For binary events a categorical contingency table can be built such as Table A1. It is often used to compute categorical
verification scores.

**Table A1.** Contingency table for a binary event. The table fully describes the joint distribution of forecast and observation for N forecast/observation pairs.

|  | Observed event | Not observed event |  |
|---|---|---|---|
| Forecast event | a | b | a + b |
| Not forecast event | c | d | c + d |
|  | a + c | b + d | N = a + b + c + d |

- The hit rate or probability of detection is defined as follows: $H = POD = \frac{a}{a+c}$

It ranges from 0 to 1 (1 for a perfect forecast).

- The false alarm rate or probability of false detection is defined as follows: $F = POFD = \frac{b}{b+d}$

It ranges from 0 to 1 (0 for a perfect forecast).

- The Heidke skill score (HSS) measures the fraction of correct forecasts after eliminating the correct random forecasts:

$HSS = 2\frac{ad-bc}{(a+c)(c+d)+(a+b)(b+d)}$



It ranges from $-\infty$ to 1 (1 for a perfect forecast).

- Frequency bias is defined as follows: $B = \frac{a+b}{a+c}$

It ranges from 0 to $\infty$.

**Gerrity score**

The Gerrity score (Gerrity Jr, 1992) is based on a multi-category contingency table such as Table A2. In PIAF algorithm, six rain thresholds: 0.05 ; 0.1 ; 0.2 ; 0.4 ; 0.8 and 1.6mm/5min are considered in the multi-category contingency table.

**Table A2.** K-category contingency table. $n(F_i,O_j)$ is the number of forecast/observed pairs for which the forecast is in category i and the observation in category j. $N(F_i)$ indicates the total number of forecasts in the category i, $N(O_j)$ indicates the total number of observations in the category j and N the total number of forecast/observed pairs.

|  | i,j | \multicolumn{4}{c}{Observed category} | Total |
|---|---|---|---|---|---|---|
|  | i,j | 1 | 2 | ... | K |  |
|  | 1 | $n(F_1,O_1)$ | $n(F_1,O_2)$ | ... | $n(F_1,O_K)$ | $N(F_1)$ |
| Forecast category | 2 | $n(F_2,O_1)$ | $n(F_2,O_2)$ | ... | $n(F_2,O_K)$ | $N(F_2)$ |
|  | ... | ... | ... | ... | ... | ... |
|  | K | $n(F_K,O_1)$ | $n(F_K,O_2)$ | ... | $n(F_K,O_K)$ | $N(F_K)$ |
| Total |  | $N(O_1)$ | $N(O_2)$ | ... | $N(O_K)$ | N |

The Gerrity score is defined as follows:

$$Gs = \frac{1}{N} \sum_{i=1}^{K} \sum_{j=1}^{K} n(F_i,O_j)s_{ij} \tag{A3}$$

where $s_{ij}$ are the elements of a matrix characterized by $s_{ii} = \frac{1}{K-1}(\sum_{r=1}^{i-1} a_r{}^{-1} + \sum_{r=i}^{K-1} a_r)$ on the diagonal and $s_{ij} = s_{ji} = \frac{1}{K-1}(\sum_{r=1}^{i-1} a_r{}^{-1} - (j-i)\sum_{r=i}^{K-1} a_r)$ elsewhere, with $a_r = \frac{1 - \sum_{t=1}^{r} p_t}{\sum_{t=1}^{r} p_t}$ and $p_t = N(O_t)/N$ the frequency of observations in category t.

It ranges from -1 to 1 (1 for a perfect forecast).



**FSS**

The Fractions Skill Score (FSS, Roberts and Lean, 2008) allows to evaluate with a certain spatial tolerance the quality of a forecast for several intensity thresholds. The idea is to compare the observed and forecast probabilities of an event in spatial windows of increasing size. The FSS is defined by:

$$FSS = 1 - \frac{\frac{1}{N}\sum_{i=1}^{N}(p_{f,i} - p_{o,i})^2}{\frac{1}{N}\left(\sum_{i=1}^{N}p_{f,i}^2 + \sum_{i=1}^{N}p_{o,i}^2\right)} \tag{A4}$$

with N the number of points in the considered spatial window, $p_{f,i}$ the forecast fraction of grid points that exceed the threshold in this window and $p_{o,i}$ the observed one.

In concrete terms, the observation and forecast fields are made binary by assigning the value 1 to pixels associated with an exceedance of the studied threshold and 0 in the opposite case. A neighborhood size is set. Then, within the neighborhood of each pixel in the area of study, the fraction of observed and forecast points above the intensity threshold is calculated.

The FSS ranges from 0 to 1, close to 1 for a good forecast and close to 0 in the opposite case. In general, as the spatial tolerance increases, the FSS increases, and as the intensity thresholds increases, the FSS decreases.

*Acknowledgements.* This research was performed in the framework of the HyMeX programme (MISTRALS grants) and is a contribution to the French project PICS (ANR – 17 – CE03 – 0011). The authors would like to express their gratitude for access to useful data and discussions with the members of the nowcasting team at Météo-France, and especially Céline Jauffret, Philippe Cau and Nicolas Merlet.



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





**FIGURES**

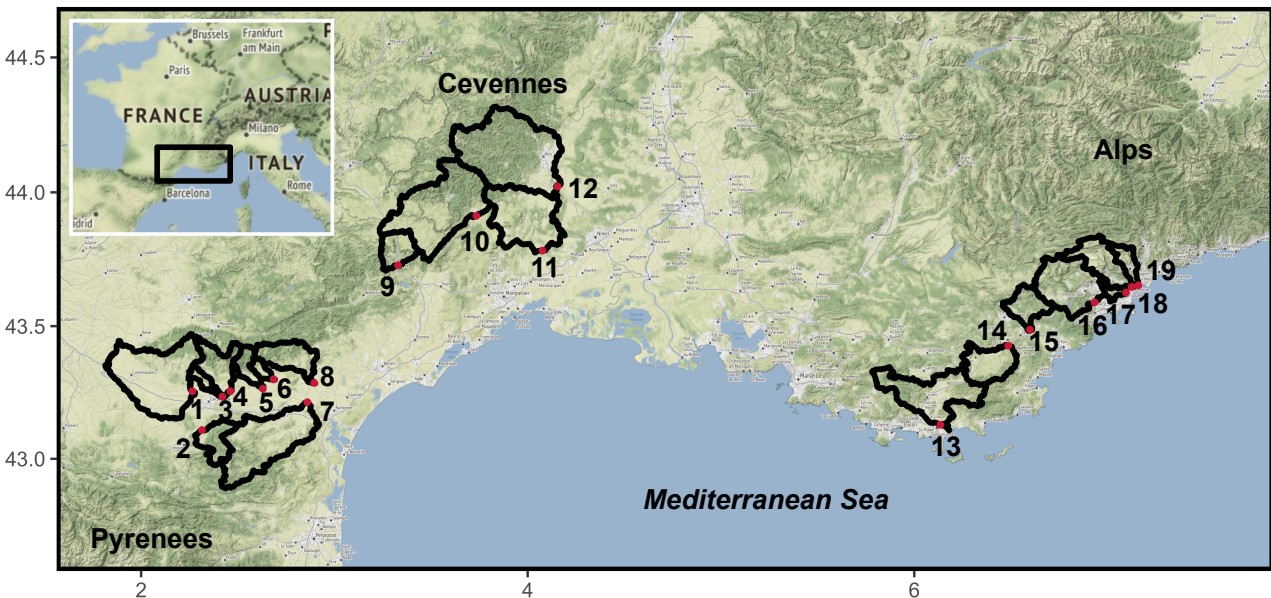

**Figure 1.** The black rectangle indicates the study area in France. The thick lines delineate the studied watersheds within this zone. The red circles correspond to the associated outlets. The base map used in the background of the plot comes from © Google Maps.

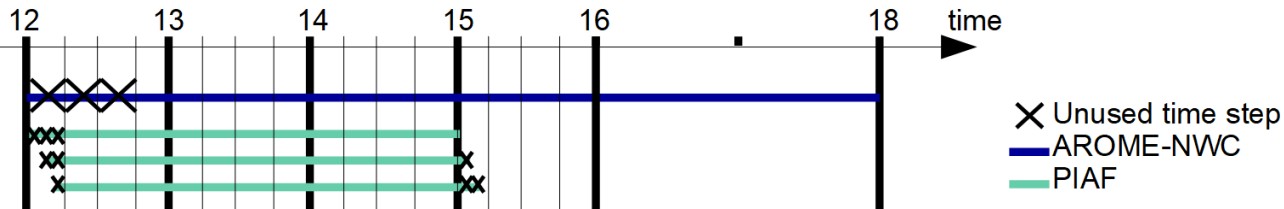

**Figure 2.** Example showing the unused time steps for a AROME-NWC forecast run starting at 1200 UTC and for three successive PIAF forecasts starting at 1200, 1205 and 1210 UTC.





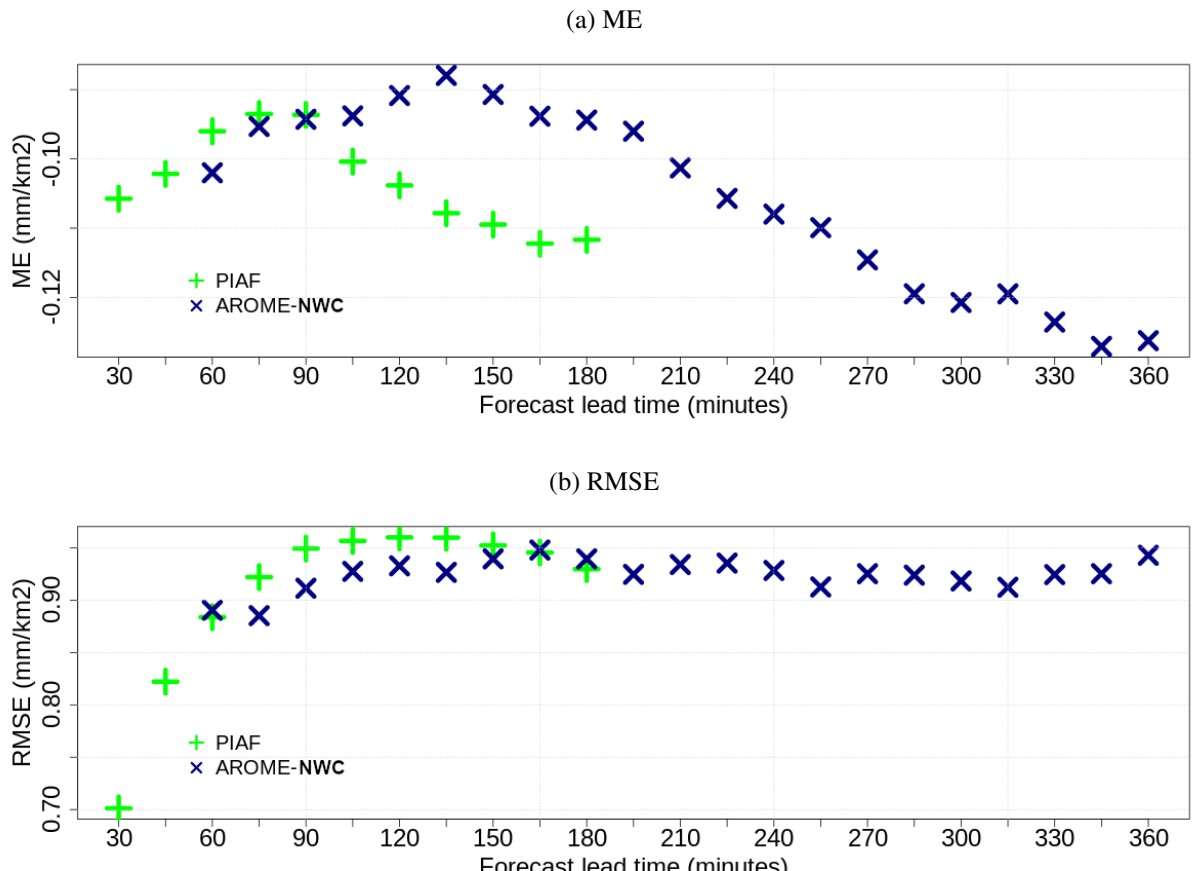

**Figure 3.** (a) Mean and (b) root mean square errors for AROME-NWC (in blue) and PIAF (in green) over their respective forecast ranges for 15-min rain accumulation forecasts. Scores computed over the box displayed Figure 1.



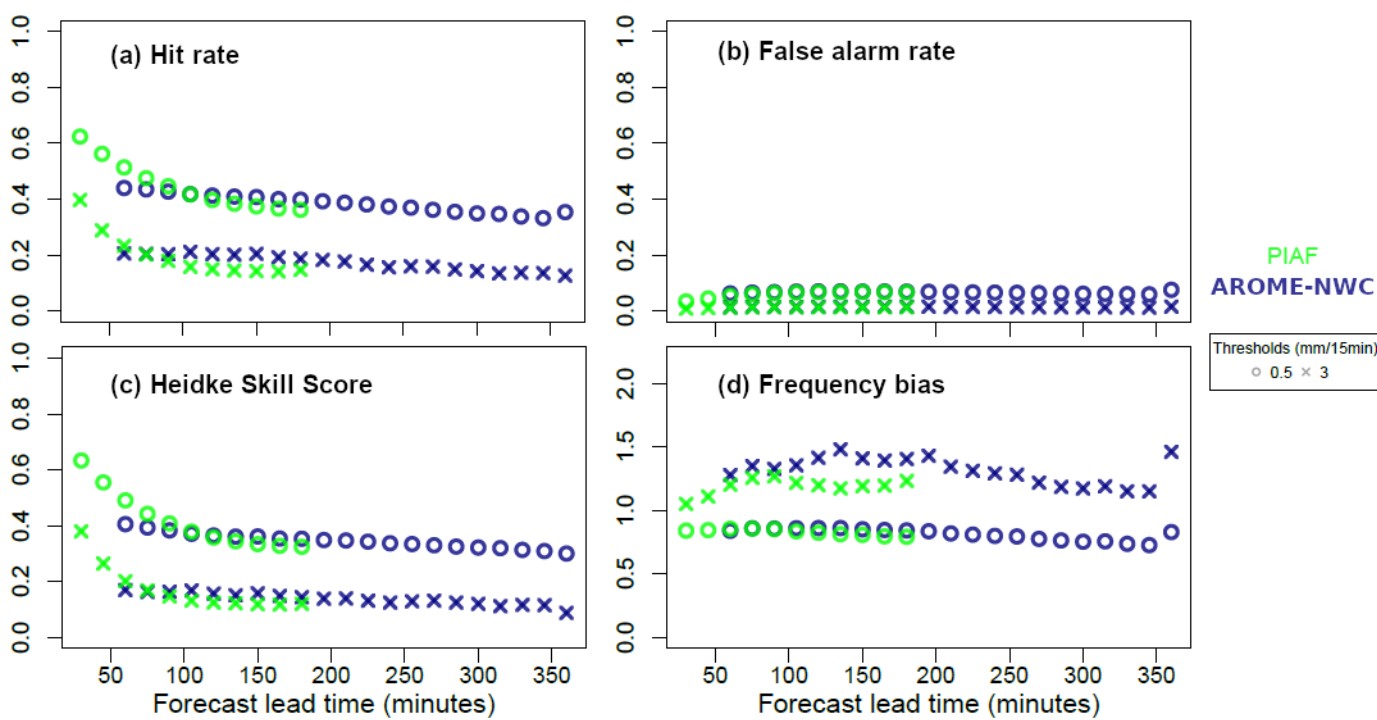

**Figure 4.** (a) Hit rate, (b) false alarm rate, (c) Heidke Skill Score, and (d) frequency bias for thresholds: 0.5mm/15min and 3mm/15min computed for AROME-NWC (in blue) and PIAF (in green) over their respective forecast ranges for 15-min rain accumulation over the box displayed Figure 1.





**Figure 5.** The 10-events mean FSS obtained with AROME-NWC for the different precipitation thresholds (0.5, 1, 2, 3, 5 et 10 mm/15min) and window sizes (1, 5, 10, 20 et 40 km) at each forecast lead time from (a) 60 min to (u) 360 min.





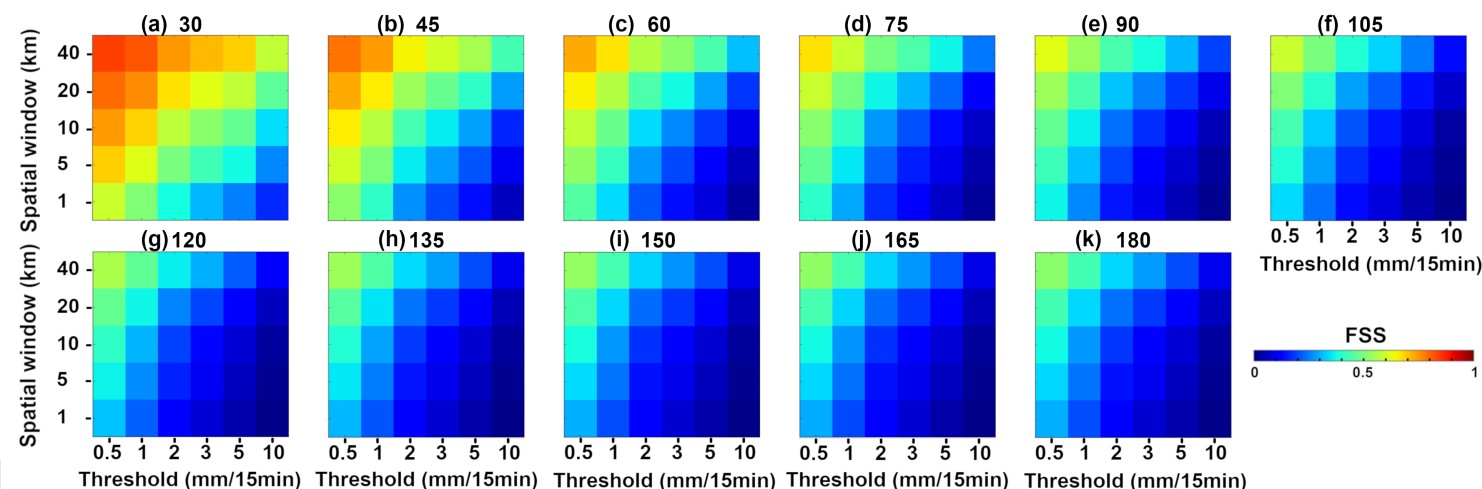

**Figure 6.** The 10-events mean FSS obtained with PIAF for the different precipitation thresholds (0.5, 1, 2, 3, 5 et 10 mm/15min) and window sizes (1, 5, 10, 20 et 40 km) at each forecast lead time from (a) 30 min to (k) 180 min.



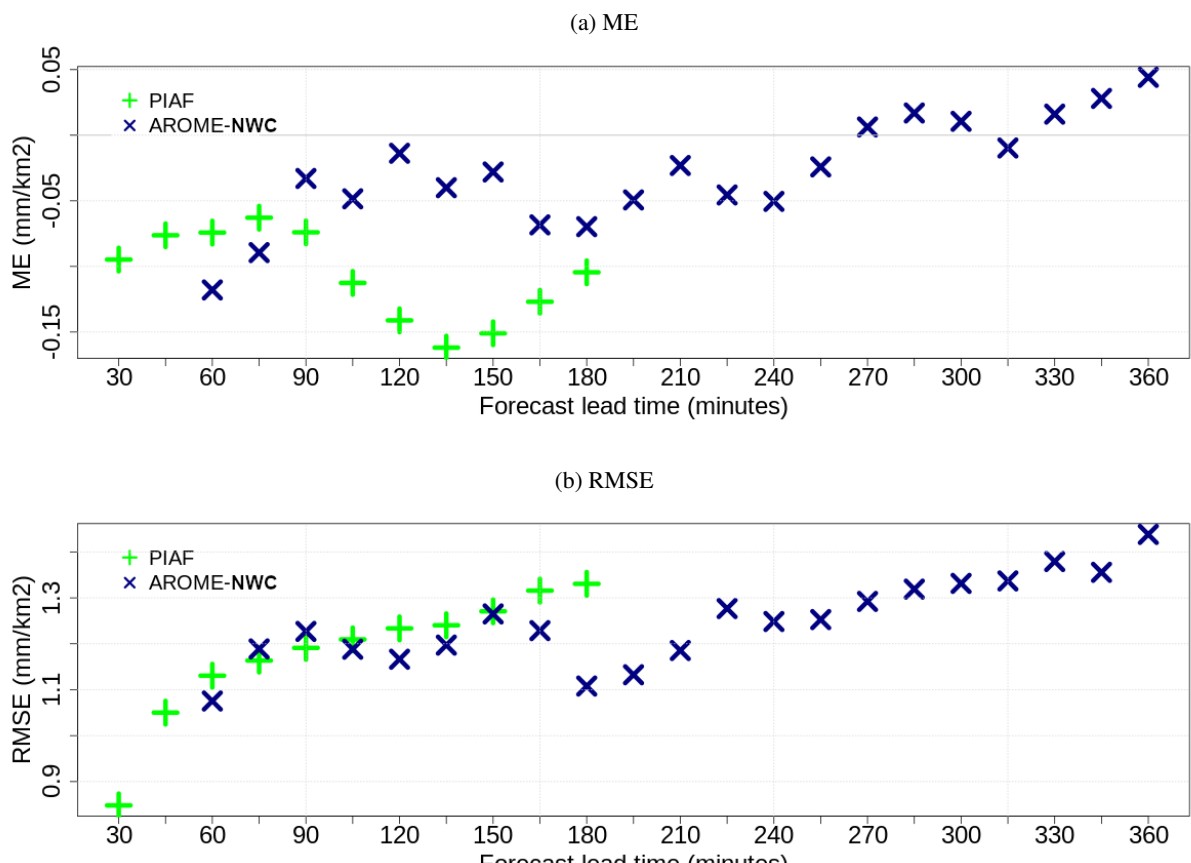

**Figure 7.** (a) Mean and (b) root mean square errors for AROME-NWC (in blue) and PIAF (in green) over their respective forecast ranges for 15-min rain accumulation forecasts averaged over the catchments.



**Figure 8.** Distributions of average rainfall forecast errors over catchment areas (forecast values - observed values) for a) AROME-PI (lead times in the range T+45 minutes to T+6 hours), and b) PIAF (lead times in the range T+15 minutes to T+3 hours) for all events and by event. The lower edge of the boxplots corresponds to the first quartile, the central line to the median and the upper edge to the third quartile of the values taken by these errors (the moustaches go up to the extreme values).



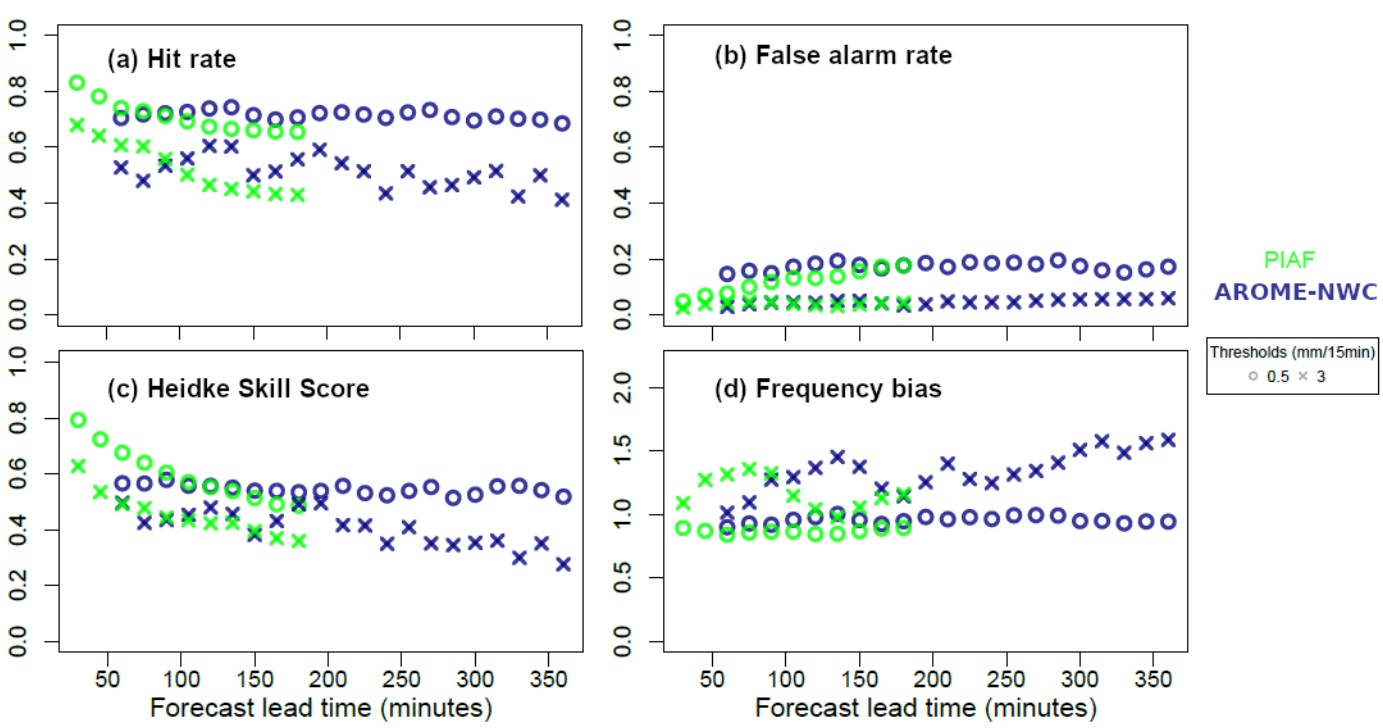

**Figure 9.** (a) Hit rate, (b) false alarm rate, (c) Heidke Skill Score, and (d) frequency bias for thresholds: 0.5mm/15min and 3mm/15min computed for AROME-NWC (in blue) and PIAF (in green) over their respective forecast ranges for 15-min rain accumulation forecasts averaged over the catchments.





**Figure 10.** Discharge time series simulated by ISBA-TOP driven by ANTILOPE (black curve), by (a) AROME-NWC and by (b) PIAF for the case 15/10/2018 for the Orbiel river at Bouilhonnac (watershed No. 4). The reverse histogram represents the ANTILOPE hourly rainfall averaged over the catchment.





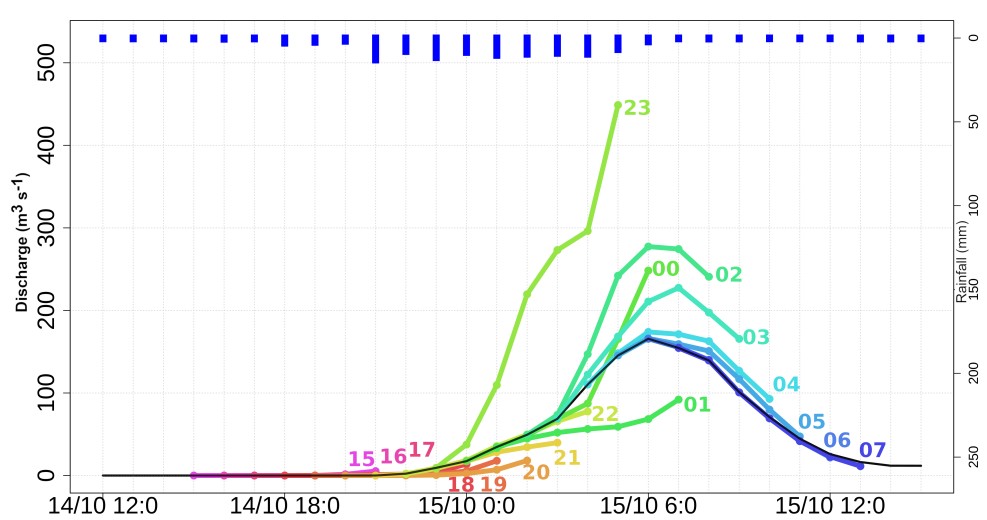

**Figure 11.** Discharge time series simulated by ISBA-TOP driven by ANTILOPE rainfall estimates (black curve) and by AROME-NWC forecasts (coloured curves with forecasts starting from 14 October 15 UTC to 15 October 07 UTC) for the Fresquel river at Pezens (watershed No. 1). The reverse histogram represents the ANTILOPE hourly rainfall averaged over the catchment.





**TABLES**

**Table 1.** Characteristics of the studied watersheds.

| River | Outlet | Area (km$^2$) | No. in Figure 1 |
|---|---|---|---|
| Fresquel | Pezens | 733 | 1 |
| Lauquet | Saint Hilaire | 173 | 2 |
| Trapel | Villedubert | 19 | 3 |
| Orbiel | Bouilhonnac | 239 | 4 |
| Argent Double | Redorte | 108 | 5 |
| Ognon | Pépieux | 47 | 6 |
| Orbieu | Villedaigne | 748 | 7 |
| Cesse | Mirepeisset | 270 | 8 |
| Lergue | Lodève | 181 | 9 |
| Hérault | Laroque | 918 | 10 |
| Vidourle | Sommières | 621 | 11 |
| Gardon | Ners | 1092 | 12 |
| Gapeau | Hyères | 548 | 13 |
| Aille | Vidauban | 279 | 14 |
| Endre | Le Muy | 187 | 15 |
| Siagne | Pégomas | 515 | 16 |
| Brague | Biot | 41 | 17 |
| Loup | Villeneuve-Loubet | 278 | 18 |
| Cagne | Cagnes-sur-Mer | 109 | 19 |



**Table 2.** Characteristics of the studied events and number of forecasts.

| Rainy period (duration) | Maximum cumulative rainfall estimated (mm) | No. of AROME-NWC forecasts | No. of PIAF forecasts | Studied watersheds |
|---|---|---|---|---|
| 3 Oct 2015 (5h) | 212 | 8 | 84 | 16,17,18,19 |
| 14 Sep 2016 (20h) | 241 | 24 | 276 | 9,10,11,12 |
| 12-14 Oct 2016 (40h) | 255 | 60 | 708 | 9,10,11,12 |
| 20-22 Nov 2016 (41h) | 303 | 14 | 156 | 13,14,15,16,17,18,19 |
| 12-14 May 2018 (46h) | 98 | 13 | 145 | 9,10,11,12 |
| 28-31 May 2018 (81h) | 233 | 67 | 781 | 9,10,11,12 |
| 11 Jun 2018 (15h) | 107 | 17 | 193 | 9,10,11,12 |
| 10-11 Oct 2018 (30h) | 234 | 30 | 348 | 13,14,15,16,17,18,19 |
| 14-15 Oct 2018 (15h) | 283 | 17 | 192 | 1,2,3,4,5,6,7,8 |
| 8-10 Nov 2018 (38h) | 213 | 27 | 311 | 9,10,11,12 |

**Table 3.** Best anticipation of the rise in flow, peak value and peak timing simulated with ISBA-TOP driven by the AROME-NWC rainfall forecast for the Fresquel river at Pezens on 14-15 October 2018. For more explanation on how to read the table, see subsection 3.3.

| Outlet (No. In Fig.1) | Discharge peaks (m³s⁻¹) | | Anticipation of the rising flow | | | | | | | Anticipation of the peak value | | | | Anticipation of the peak timing | | | | |
|---|---|---|---|---|---|---|---|---|---|---|---|---|---|---|---|---|---|---|
| | observed | reference | | H-5 | H-4 | H-3 | H-2 | H-1 | H | H-2 | H-1 | H | | H-4 | H-3 | H-2 | H-1 | H |
| Pezens (1) | 173 | 166 | 5/4 | | | | | | | 4/4/2 | | | | | | | | |





**AROME-NWC**

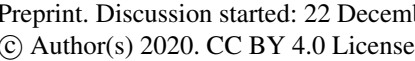

| | | Discharge peaks (m³s⁻¹) | | Anticipation of the rising flow | Anticipation of the peak value | Anticipation of the peak timing |
|---|---|---|---|---|---|---|
| | Outlet (No. In Fig.1) | observed | reference | H-7 | H-5 | |
| 03/10/15 | Pégomas (16) | 215 | 214 | 4/4 | 6/4/4 | |
| | Biot (17) | 240 | 275 | 4/4 | 6/6/1 | |
| | Villeneuve-Loubet (18) | 127 | 160 | 4/4 | 6/6/5 | |
| 15/10/18 | Pezens (1) | 173 | 166 | 5/4 | 4/4/2 | |
| | Saint-Hilaire (2) | 238 | 190 | 7/7 | 3/3/3 | |
| | Villedubert (3) | ? | 298 | 5/4 | 4/4/2 | |
| | Bouilhonnac (4) | 481 | 414 | 7/6 | 4/4/3 | |
| | Redorte (5) | 169 | 129 | 9/3 | 7/6/4 | |
| | Pépieux (6) | 77 | 45 | 9/4 | 6/3/4 | |
| | Villedaigne (7) | 1000 | 363 | 4/4 | 5/5/2 | |
| | Mirepeisset (8) | 574 | 293 | 6/4 | 4/2/2 | |

**Table 4.** Best anticipation of the rise in flow, peak value and peak timing simulated with ISBA-TOP driven by the AROME-NWC rainfall forecast for the watersheds affected during the events of October 2015 and 2018. The green colour (respectively orange, red) indicates an intensity error on the forecast peak lower than 10% (respectively 20%, 30%) compared to the reference peak. Streamflow observations which were provided by the French HYDRO data bank (http://www.hydro.eaufrance.fr/, last access: 11 March 2020) or by HyMeX post-event surveys are for information purposes only. For more explanation on how to read the table, see subsection 3.3.



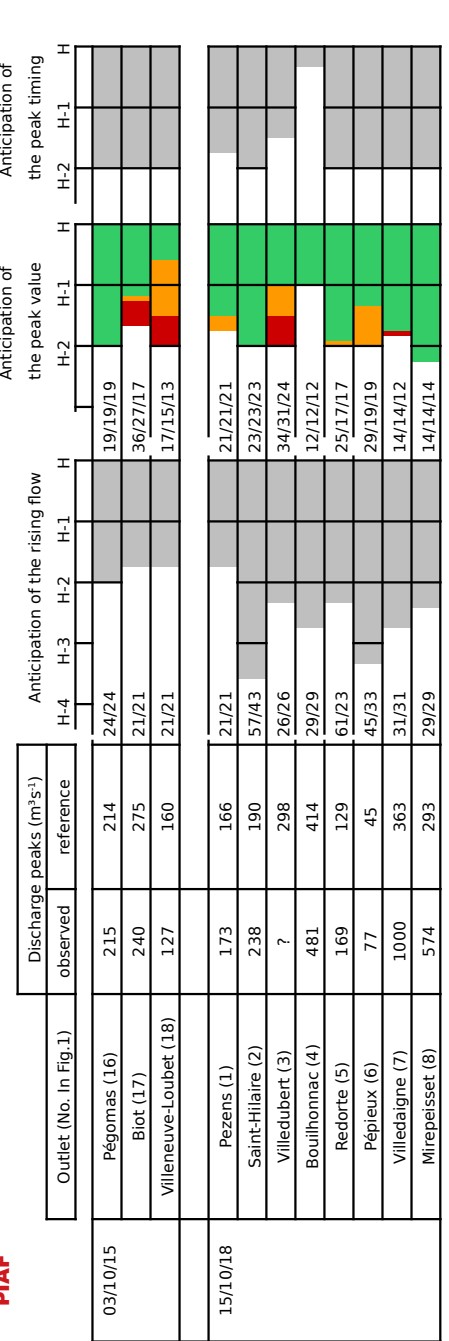

**Table 5.** Best anticipation of the rise in flow, peak value and peak timing simulated with ISBA-TOP driven by the PIAF rainfall forecast for the watersheds affected during the events of October 2015 and 2018. The green colour (respectively orange, red) indicates an intensity error on the forecast peak lower than 10% (respectively 20%, 30%) compared to the reference peak. Streamflow observations which were provided by the French HYDRO data bank (http://www.hydro.eaufrance.fr/, last access: 11 March 2020) or by HyMeX post-event surveys are for information purposes only. For more explanation on how to read the table, see subsection 3.3.