# Peer review of "Hydrometeorological evaluation of two nowcasting systems for Mediterranean heavy precipitation events with operational considerations"

_Hydrology and Earth System Sciences, 2020_

## Referee Comment (RC1) · Anonymous Referee #1 · 20 Jan 2021

The paper describes the evaluation of the performances of 2 nowcasting systems both in terms of precipitation and of hydrological response and anticipation. The two systems were analyzed and tested on an area in the south of France considering various basins in the hydrological analysis.

I think the paper is interesting and quite well structured even if in some parts difficult to read. The topic of the paper is of scientific and operational interest.

I suggest to accept the paper after some modifications.

Major issues:

Section 2.2.2. I think the authors should be more detailed about PIAF, I see they cite another paper and a presentation (found on web) but this is an interesting part of the paper. They should insert more formalization about PIAF about the blending methods and some equations and if necessary a scheme, so that readers can understand how the system works. Currently the paragraph is too generic.

It is not clear to me why the authors did not evaluate the products on the same time window. I understand that probably PIAF from a certain lead time head will tend to be really similar to AROME-NWC (or not?). Looking at figures 3, 4, 7 (and others) I would found interesting having the lines (or dots) on same time windows (60 to 360 min). Since I have some experience about how is sometimes (potentially) hard and time consuming a request like this, I do not want to constrain the authors to enlengthen all the PIAF graphs if they have not the run easily available. Anyway they should insert some discussion about this point.

Section 3.3 Hydrological verification is interesting, but I have to say is quite complex to follow in some passages. The explanation is detailed but sometime I loose myself between reading it and looking the figures ore the tables. I suggest inserting bullet points along the explanation in order to make more schematic the explanations. Two elements would be interesting:

1) As I said before for rain evaluation, it would be interesting the case of using the blending in PIAF algorithm in range 60 360 minutes, and trigger hydro model. I do not know if this is possible at least for the two events of hydrological evaluation; if this request too work I do not ask to the authors to carry out the analysis.
2) The propagation capability of hydro model. The authors states "The evaluation aims at addressing the following question: How many hours of anticipation on floods can we have at most in terms of intensity and temporality of the flood peaks using rainfall nowcasting". Hydro model could have a key role a in some cases. I believe it would be interesting to see the graphs in Figure 10 and 11, leaving the routing of hydro model working for 3-6 hours. It seems to me that hydro model is not exploited in all its potentialities. This reflects also in the Conlusions.

Other comments:

Pgg 2 lines 25-29, the sentence is not clear, pleas rephrase. Line 29 maybe observations not observation.

Pgg 3 lines 77. Not clear . do you mean "verification methods"?

Pgg 6, line 168 "….The reference is the discharge simulation obtained using the radar rainfall estimates ANTILOPE as input to the distributed hydrological.." this is a reasonable ad used approach, I only suggest to insert some refereces

Pgg 7, line 199 "…PIAF results from the linear combination of AROME-NWC prediction fields and radar extrapolation. The weights given to each predictor are adjusted according to their recent performance against observation…." It is not clear in which way, more details in section 2.2.2 e some more comments here can help to understand.

Pgg 8, lines 227 231. Please rephrase this paragraph is not really clear. I would start "In order to veify…"

Pgg 9 lines 254-255. Please rephrase

Pgg 10 lines 305. Remove "the" in sentence .."the four…"

Pgg 10 lines 320 – 321. "Anticipation time is not proportional to the size of the watersheds, i.e. the smaller catchments with the lowest concentration times do not necessarily have a shorter anticipation time".

I believe that this depends on the single events, and how much it is well forecast in terms of rainfall which is the driver of hydro model. Anyway in some cases this could be also due by the fact that authors stopped the simulations with rainfall availability. It could be interesting to evaluate how results changes exploiting the propagation capability of the hydro model, if and when a gain in anticipation of the flood is obtained.

Pgg 13 the range of RMSE is [0, infinite]

Table 4: outlet Repieux, it is not clear to me if the sequence 5/3/4 is possible. But again

---

## Referee Comment (RC2) · Anonymous Referee #2 · 10 Jun 2021

hess-2020-629 review: Hydrometeorological evaluation of two nowcasting systems for Mediterranean heavy precipitation events with operational considerations

The purpose of this article is to compare two different meteorological nowcasting products AROME-NWC and PIAF in South-eastern France. The study is conducted both from a meteorological point of view (comparison of cumulated rainfall on the whole domain) and from a hydrological point of view (comparison of cumulated rainfall at the catchment scale and corresponding discharges simulated with the hydrological model

ISBA-TOP).

The topic is of great interest in the field of hydrology and the article clearly shows the potential of these nowcasting products for Mediterranean events. However, the present article lacks a global view on two main points:

- The added value of these nowcasting systems with respect to traditional forecasting systems for Mediterranean heavy precipitation events: only nowcasting systems have been tested without any comparison or analysis of other existing systems,

- The forecasted discharges are compared to reference discharges, simulated with observation of precipitations: this makes it possible not to take into account the uncertainties in the model structure and parametrization but adds the uncertainties related to precipitation observations. It would have been interesting to extend the analysis also to observed discharges and see to what extent the current conclusions are still valid.

From the results and analysis, it also seems to me that AROME-NWC is more promising than PIAF for flash-flood forecasting, at least on the tested events and catchments, yet it is not clearly stated either on the abstract, or in the conclusions. Am I missing something on the added value of PIAF on that point?

1. P4 L119: what is the "regret"? A more detailed description of PIAF will be interesting for a better understanding of the results and analysis, without the need to read several other publications.

2. P5 L123: the Gerrity score is detailed in Appendix A, you should add a cross-reference here for clarification.

3. P5 §2.3: how is handled the different spatial resolution between ISBA (300m) and TOPODYN (50m)?

4. P6 L171: how is ISBA-TOP calibrated? Using ANTILOPE rainfall estimates and observed discharges at the catchment outlet? With continuous or event-based simulations? On which time period?

5. P7 L186: AROME-NWC shows a trend to predict too frequently high rainfall accumulation but at the same time precipitations are underestimated by the model on average (see mean error figure 3). I'm not sure I correctly get this point: does it indicate a questionable representation of the dynamic of precipitation (high peaks forecasted instead of continuous precipitation of lower intensity)?

6. Table 2, second column: maximum cumulative rainfall estimate (mm): where does this estimation come from? ANTILOPE radar product?

---

## Author Comment (AC1) · 8 Sep 2021

POINT BY POINT ANSWERS TO REFEREE 1

**The paper describes the evaluation of the performances of 2 nowcasting systems both in terms of precipitation and of hydrological response and anticipation. The two systems were analyzed and tested on an area in the south of France considering various basins in the hydrological analysis.**

**I think the paper is interesting and quite well structured even if in some parts difficult to read. The topic of the paper is of scientific and operational interest.**

**I suggest to accept the paper after some modifications.**

⇨ We would like to express our gratitude towards the time that Referee1 dedicated to providing valuable feedback and suggestions to help in improving this paper. In the following, we reply to each comment and indicate how the suggestions will be taken into account in the new version of the manuscript.

**Major issues:**

**Section 2.2.2. I think the authors should be more detailed about PIAF, I see they cite another paper and a presentation (found on web) but this is an interesting part of the paper. They should insert more formalization about PIAF about the blending methods and some equations and if necessary a scheme, so that readers can understand how the system works. Currently the paragraph is too generic.**

⇨ In the revised manuscript, we will expand this section by including more details about PIAF. The second paragraph will be modified and an additional figure will be added as follows:

"PIAF is based on a sequential aggregation of these two predictors (radar extrapolation and numerical prediction) and the results of blending is a linear compound of both of the form: PIAF =α * Extrapolation + (1-α) * AROME-NWC. Its aim is to perform better than the best predictor. The accuracy of a prediction proposed by the experts (radar extrapolation and AROME-NWC) or by PIAF is measured through a loss function. The Gerrity score (Gerrity Jr, 1992) described in Appendix A is here used to estimate the loss of each product with respect to the radar quantitative precipitation estimates. The difference between the forecaster's accumulated loss and that of an expert is called regret, as it measures how much the forecaster regrets, in hindsight, of not having followed the advice of this particular expert (Cesa-Bianchi and Lugosi 2006). As the forecaster's goal is to minimize the regret, the weights given to each predictor in PIAF are adjusted according to their deviation from the previous 6 hours observations, this results in weighting more the expert whose cumulative loss is small. The polynomially weighted average forecaster with multiple learning rates (ML-Poly, Cesa-Bianchi and Lugosi 2006, Gaillard and Goude, 2015) is the aggregation rule used in PIAF to assign weights to each predictor. This method provides a real choice of predictor rather than a mixture. The weights depend also on the forecast range (additional Figure) and on the geographical area, according to a division of France into six sub-areas. PIAF is run every 5 minutes with a 3 hours lead time and a time step of 5 minutes. Forecasts are available within 2 minutes."

Cesa-Bianchi, N., & Lugosi, G. (2006). *Prediction, learning, and games*. Cambridge university press.

Gaillard, P., & Goude, Y. (2015). Forecasting electricity consumption by aggregating experts; how to design a good set of experts. In *Modeling and stochastic learning for forecasting in high dimensions* (pp. 95-115). Springer, Cham.

[Figure]

*Additional figure : 3-D representation of the weight α given to radar extrapolation in PIAF. It shows the PIAF forecast lead time (interval [0, 180 minutes]) dependency on α (interval [0, 1]) for PIAF forecasts starting from 12 October 2016 18:05UTC to 13 October 2016 00:05UTC.*

**It is not clear to me why the authors did not evaluate the products on the same time window. I understand that probably PIAF from a certain lead time head will tend to be really similar to AROME-NWC (or not?). Looking at figures 3, 4, 7 (and others) I would found interesting having the lines (or dots) on same time windows (60 to 360 min). Since I have some experience about how is sometimes (potentially) hard and time consuming a request like this, I do not want to constrain the authors to enlengthen all the PIAF graphs if they have not the run easily available. Anyway they should insert some discussion about this point.**

⇨ The products were not evaluated on the same time window because their lead times and forecast intervals are different. The nowcasting system PIAF has been developed to provide forecasts up to 3 hours and thus ensure the transition to the "classical" forecasting lead times.

New sentences Pgg 6 line 164 in section 2.4 will be added to emphasise this point:

"As PIAF forecasts last 180 minutes and AROME-NWC forecasts last 360 minutes, they are not evaluated on the same time window. Their performance can be compared only at lead times less than 180 minutes"

**Section 3.3 Hydrological verification is interesting, but I have to say is quite complex to follow in some passages. The explanation is detailed but sometime I loose myself between reading it and looking the figures ore the tables. I suggest inserting bullet points along the explanation in order to make more schematic the explanations.**

⇨ Thank you for the suggestion. Bullet points will be inserted along the explanation in the revised manuscript.

**Two elements would be interesting:**

**1) As I said before for rain evaluation, it would be interesting the case of using the blending in PIAF algorithm in range 60 360 minutes, and trigger hydro model. I do not know if this is possible at least for the two events of hydrological evaluation; if this request too work I do not ask to the authors to carry out the analysis.**

**2) The propagation capability of hydro model. The authors states "The evaluation aims at addressing the following question: How many hours of anticipation on floods can we have at most in terms of intensity and temporality of the flood peaks using rainfall nowcasting". Hydro model could have a key role a in some cases. I believe it would be interesting to see the graphs in Figure 10 and 11, leaving the routing of hydro model working for 3-6 hours. It seems to me that hydro model is not exploited in all its potentialities. This reflects also in the Conlusions.**

⇨ 1) As operational PIAF forecasts last only 180 minutes, the hydrological evaluation was based on this time range.

⇨ 2) In this paper we have focused on the contribution of rainfall forecasts only and not on the impact of the propagation capability of ISBA-TOP. Our main objective was to assess the quality of the rainfall forecasts rather than the quality of the hydrological model. However we totally agree that the issue of the propagation capability of hydrological models needs to be investigated further in other studies including nowcasting ranges. The end of the Conclusions will be modified to reflect this aspect:

Pgg 12 line 384 "...Up to one hour and a half, or two hours, PIAF allows to forecast well peak value and time. Even if this hydrological assessment of the nowcasting systems provides positive and encouraging results for flash-flood forecasting, simulations could certainly be improved by using the propagation capability of the hydrological model. By letting the hydrological model route the water few hours after the end of the rainfall forecast, anticipation times could be increased. This point could be addressed within the framework of the French PICS project (Payrastre et al., 2019) as well as the use of other hydrological models and new case studies for further analysis."

**Other comments:**

**Pgg 2 lines 25-29, the sentence is not clear, pleas rephrase. Line 29 maybe observations not observation.**

⇨ The sentences will be modified as follows:

"It is difficult to forecast heavy precipitation events with accurate intensity, chronology and location. Among the difficulties encountered there are the complex features and variability of deep convection and the associated small space-time scales that are hardly predictable. Nowcasting systems suit these scales with high spatial and temporal resolution short-term forecasts (usually up to a few hours). They can be based on extrapolation of observations, or rely on mesoscale numerical weather prediction or else combine these two approaches."

**Pgg 3 lines 77. Not clear. do you mean "verification methods"?**

⇨ Yes we mean "verification methods". All "evaluation methods" will be replaced by "verification methods" in the revised manuscript.

**Pgg 6, line 168 "....The reference is the discharge simulation obtained using the radar rainfall estimates ANTILOPE as input to the distributed hydrological.." this is a reasonable ad used approach, I only suggest to insert some references**

⇨ A new sentence with three references will be added:

"...The reference is the discharge simulation obtained using the radar rainfall estimates ANTILOPE as input to the distributed hydrological model. This approach allows to dissociate the error made by the hydrological model from that made by the rainfall forecasts (Borga, 2002, Berenguer et al., 2005, Poletti et al. 2019)."

Borga, M. (2002). Accuracy of radar rainfall estimates for streamflow simulation. *Journal of Hydrology*, *267*(1-2), 26-39.

Berenguer, M., Corral, C., Sánchez-Diezma, R., & Sempere-Torres, D. (2005). Hydrological validation of a radar-based nowcasting technique. *Journal of Hydrometeorology*, *6*(4), 532-549.

Poletti, M. L., Silvestro, F., Davolio, S., Pignone, F., & Rebora, N. (2019). Using nowcasting technique and data assimilation in a meteorological model to improve very short range hydrological forecasts. *Hydrology and Earth System Sciences*, *23*(9), 3823-3841.

**Pgg 7, line 199 "...PIAF results from the linear combination of AROME-NWC prediction fields and radar extrapolation. The weights given to each predictor are adjusted according to their recent performance against observation...." It is not clear in which way, more details in section 2.2.2 e some more comments here can help to understand.**

⇨ We hope that the details provided in Section 2.2.2 will make the analysis of the results clearer. The sentence Pgg7 line 199 will be modified as follows:

"The weights given to each predictor are adjusted according to their recent performance against observation as described in section 2.2.2."

**Pgg 8, lines 227 231. Please rephrase this paragraph is not really clear. I would start "In order to verify..."**

⇨ The paragraph in the new version of the paper will be modified as follows:

"In order to verify the forecast performance of AROME-NWC and PIAF, two verification methods were used: a traditional point-to-point verification and a neighborhood spatial technique using FSS. The results obtained with these two methods are similar and can be summarized as follows: a quick loss of PIAF accuracy is observed on the very first lead times, its performance is higher than AROME-NWC up to 1h15/1h30 of forecast but not necessarily beyond."

**Pgg 9 lines 254-255. Please rephrase**

⇨ This will be rephrased as follows:

"Finally, the results of both verification methods (point-to-point and catchment scale comparisons of observed and forecast rainfall) generally lead to the same conclusions."

**Pgg 10 lines 305. Remove "the" in sentence .."the four…"**

⇨  It will be removed in the revised manuscript.

**Pgg 10 lines 320 – 321. "Anticipation time is not proportional to the size of the watersheds, i.e. the smaller catchments with the lowest concentration times do not necessarily have a shorter anticipation time".**

**I believe that this depends on the single events, and how much it is well forecast in terms of rainfall which is the driver of hydro model. Anyway in some cases this could be also due by the fact that authors stopped the simulations with rainfall availability. It could be interesting to evaluate how results changes exploiting the propagation capability of the hydro model, if and when a gain in anticipation of the flood is obtained.**

⇨ This remark is relevant and it is clear that studying the propagation capability of the hydrological model would be very interesting, even if in this study we have focused more on the contribution of the rainfall forecasts alone to provide useful information.

**Pgg 13 the range of RMSE is [0, infinite]**

⇨ Thank you for seeing it, it will be modified in the revised manuscript.

**Table 4: outlet Pépieux, it is not clear to me if the sequence 6/3/4 is possible. But again**

⇨ You are right, there was a typing error, the correct sequence is 6/3/3.
"6" for the forecasts starting at 19UTC, 21UTC, 02UTC, 03UTC, 04UTC and 05UTC (Figure 3 hereafter). "3" for the forecasts starting at 02UTC, 04UTC and 05UTC.

[Figure]

*Figure 3: Discharge time series simulated by ISBA-TOP driven by ANTILOPE rainfall estimates (black curve) and by AROME-NWC forecasts (coloured curves with forecasts starting from 14 October 15 UTC to 15 October 07 UTC) for the Ognon river at Pépieux. The reverse histogram represents the ANTILOPE hourly rainfall averaged over the catchment*

---

## Author Comment (AC2) · 8 Sep 2021

ANSWERS TO THE REFEREES FOR THE ARTICLE ENTITLED
"HYDROMETEOROLOGICAL EVALUATION OF TWO NOWCASTING SYSTEMS FOR
MEDITERRANEAN HEAVY PRECIPITATION EVENTS WITH OPERATIONAL CONSIDERATIONS."
SUBMITTED TO HYDROLOGY AND EARTH SYSTEM SCIENCES

POINT BY POINT ANSWERS TO REFEREE 2

⇨ We would like to thank Referee2 for the time she/he spent on our manuscript and for the useful and constructive comments that will help to improve the quality of the manuscript. In the following, we will answer each comment and indicate how the suggestions have been taken into account in the new version of the manuscript.

**hess-2020-629 review: Hydrometeorological evaluation of two nowcasting systems for Mediterranean heavy precipitation events with operational considerations**

**The purpose of this article is to compare two different meteorological nowcasting products AROME-NWC and PIAF in South-eastern France. The study is conducted both from a meteorological point of view (comparison of cumulated rainfall on the whole domain) and from a hydrological point of view (comparison of cumulated rainfall at the catchment scale and corresponding discharges simulated with the hydrological model ISBA-TOP).**

**The topic is of great interest in the field of hydrology and the article clearly shows the potential of these nowcasting products for Mediterranean events. However, the present article lacks a global view on two main points:**
**- The added value of these nowcasting systems with respect to traditional forecasting systems for Mediterranean heavy precipitation events: only nowcasting systems have been tested without any comparison or analysis of other existing systems,**

⇨ In this study nowcasting systems have been tested without any comparison or analysis of other existing systems for several reasons:
- Precipitation forecasts were evaluated here at a 15 min time resolution whereas most of traditional forecasting systems are run with longer time steps (usually one hour).
- The availability times of rainfall forecasts are taken into account in this study to consider the operational real time constraints. Traditional forecasts are not necessarily quickly delivered. For instance, the effective availability time of the forecasts of AROME-France (which is the convective-scale numerical weather prediction system running operationally at Météo-France) varies from 2h45 to 5h05 depending on the starting time of the forecast. At the nowcasting ranges (few minutes to 6h), this delay is not negligible.
- Nowcasting systems bridge the gap with the "classical" forecasting lead times. They should be used in addition to traditional forecasting systems.

**- The forecasted discharges are compared to reference discharges, simulated with observation of precipitations: this makes it possible not to take into account the uncertainties in the model structure and parametrization but adds the uncertainties related to precipitation observations. It would have been interesting to extend the analysis also to observed discharges and see to what extent the current conclusions are still valid.**

⇨ Comparing forecasted discharges with reference discharges is a common practice to assess rainfall forecasts quality. A new sentence with three references will be added in the revised manuscript:

"...The reference is the discharge simulation obtained using the radar rainfall estimates ANTILOPE as input to the distributed hydrological model. This approach allows to dissociate the error made by the hydrological model from that made by the rainfall forecasts (Borga, 2002, Berenguer et al., 2005, Poletti et al. 2019)."

Borga, M. (2002). Accuracy of radar rainfall estimates for streamflow simulation. *Journal of Hydrology*, *267*(1-2), 26-39.

Berenguer, M., Corral, C., Sánchez-Diezma, R., & Sempere-Torres, D. (2005). Hydrological validation of a radar-based nowcasting technique. *Journal of Hydrometeorology*, *6*(4), 532-549.

Poletti, M. L., Silvestro, F., Davolio, S., Pignone, F., & Rebora, N. (2019). Using nowcasting technique and data assimilation in a meteorological model to improve very short range hydrological forecasts. *Hydrology and Earth System Sciences*, *23*(9), 3823-3841.

**From the results and analysis, it also seems to me that AROME-NWC is more promising than PIAF for flash-flood forecasting, at least on the tested events and catchments, yet it is not clearly stated either on the abstract, or in the conclusions. Am I missing something on the added value of PIAF on that point?**

⇨ We would say that both provide valuable information but on different lead times. AROME-NWC forecasts are appropriate for lead times greater than two hours and PIAF forecasts for very first lead times. The availability time of the PIAF rainfall forecasts and their frequency might be very useful for planning the intervention of emergency services in crisis time.

**1. P4 L119: what is the "regret"? A more detailed description of PIAF will be interesting for a better understanding of the results and analysis, without the need to read several other publications.**

⇨ In the revised manuscript, we will expand section 2.2.2 by including more details about PIAF. The second paragraph will be modified and an additional figure will be added as follows:

"PIAF is based on a sequential aggregation of these two predictors (radar extrapolation and numerical prediction) and the results of blending is a linear compound of both of the form: PIAF =$\alpha$ * Extrapolation + (1-$\alpha$) * AROME-NWC. Its aim is to perform better than the best predictor. The accuracy of a prediction proposed by the experts (radar extrapolation and AROME-NWC) or by PIAF is measured through a loss function. The Gerrity score (Gerrity Jr, 1992) described in Appendix A is here used to estimate the loss of each product with respect to the radar quantitative precipitation estimates. The difference between the forecaster's accumulated loss and that of an expert is called regret, as it measures how much the forecaster regrets, in hindsight, of not having followed the advice of this particular expert (Cesa-Bianchi and Lugosi 2006). As the forecaster's goal is to minimize the regret, the weights given to each predictor in PIAF are adjusted according to their deviation from the previous 6 hours observations, this results in weighting more the expert whose cumulative loss is small. The polynomially weighted average forecaster with multiple learning rates (ML-Poly, Cesa-Bianchi and Lugosi 2006, Gaillard and Goude, 2015) is the aggregation rule used in PIAF to assign weights to each predictor. This method provides a real choice of predictor rather than a mixture. The weights depend also on the forecast range (additional

Figure) and on the geographical area, according to a division of France into six sub-areas. PIAF is run every 5 minutes with a 3 hours lead time and a time step of 5 minutes. Forecasts are available within 2 minutes."

Cesa-Bianchi, N., & Lugosi, G. (2006). *Prediction, learning, and games*. Cambridge university press.

Gaillard, P., & Goude, Y. (2015). Forecasting electricity consumption by aggregating experts; how to design a good set of experts. In *Modeling and stochastic learning for forecasting in high dimensions* (pp. 95-115). Springer, Cham.

[Figure]

*Additional figure : 3-D representation of the weight α given to radar extrapolation in PIAF. It shows the PIAF forecast lead time (interval [0, 180 minutes]) dependency on α (interval [0, 1]) for PIAF forecasts starting from 12 October 2016 18:05UTC to 13 October 2016 00:05UTC.*

**2. P5 L123: the Gerrity score is detailed in Appendix A, you should add a crossreference here for clarification.**

⇨ A crossreference will be added in the revised manuscript.

**3. P5 §2.3: how is handled the different spatial resolution between ISBA (300m) and TOPODYN (50m)?**

⇨ To answer this question and to make it clearer in the revised manuscript, the ISBA-TOP section (2.3) will be modified as follows: (P5 L129)

"This coupling consists in introducing into ISBA a lateral distribution of soil water following TOPODYN concept. ISBA deals with the water and energy budgets within the soil column and between the vegetation and the atmosphere above. Fluxes are computed for all grid meshes of its domain. From the resulting volumetric water content over a ISBA grid cell, water-storage deficit as well as the hill slope recharge are determined on the corresponding TOPODYN watershed pixels of 50-m x 50-m resolution. TOPODYN manages the computation of the lateral redistribution of water within the catchment by using topographical indexes and the spatial variability of the rainfall. The

new saturated areas and new soil moisture fields obtained by TOPODYN are then aggregated on the ISBA mesh to update water contents in ISBA. From them, ISBA computes sub-surface runoff and deep drainage which are dispached on each 50m-sided pixel and then routed up to the river and total discharges are then produced at catchment outlets."

**4. P6 L171: how is ISBA-TOP calibrated? Using ANTILOPE rainfall estimates and observed discharges at the catchment outlet? With continuous or event-based simulations? On which time period?**

⇨ In ISBA-TOP, most of the soil hydrodynamic parameters can be derived from soil data through PedoTransfer Function approaches. Other parameters such as speeds of transfer on the hillslopes and in the river can be calibrated. In this study, default values fitting Mediterranean watersheds discharge simulations are used. An example of the calibration of ISBA-TOP parameters for the simulation of flash-flood events can be found in Bouilloud et al. (2010) on three watersheds located in the French Cévennes–Vivarais region. Ten events that have occurred between 2000 and 2005 were used for the calibration and the independent evaluation. The calibration is based on the comparison of observed discharges time series at the catchment outlet with discharge time series simulated by ISBA-TOP driven by 1-h accumulated rainfall fields from rain gauges spatially interpolated with a kriging method.

The sentence P6 line 171 will be modified as follows:

"ISBA-TOP, which needs calibration for its routing parameters, was calibrated as described by Bouilloud et al., 2010 for hourly rainfall estimates, thus only hourly discharges were simulated."

**5. P7 L186: AROME-NWC shows a trend to predict too frequently high rainfall accumulation but at the same time precipitations are underestimated by the model on average (see mean error figure 3). I'm not sure I correctly get this point: does it indicate a questionable representation of the dynamic of precipitation (high peaks forecasted instead of continuous precipitation of lower intensity)?**

⇨ It is true that considering AROME-NWC forecasts at a 15 min time resolution high rainfall accumulation are predicted too frequently and precipitations are underestimated on average according to the point-to-point comparisons of the forecasts and observations. This verification based on short time steps gives significant weight to even small timing errors, especially in the case of convective situations. That's why it seems difficult to really conclude on the representation of the dynamic of precipitation with AROME-NWC. Furthermore, these results need to be considered with caution because the sample of events and catchments was limited.

**6. Table 2, second column: maximum cumulative rainfall estimate (mm): where does this estimation come from? ANTILOPE radar product?**

⇨ You are right the second column was filled in using the quantitative precipitation estimates from ANTILOPE.